# Host factor Rab11a is critical for efficient assembly of influenza A virus genomic segments

**Julianna Han**[1], **Ketaki Ganti**[2], **Veeresh Kumar Sali**[3], **Carly Twigg**[3], **Yifeng Zhang**[3], **Senthamizharasi Manivasagam**[3], **Chieh-Yu Liang**[3], **Olivia A. Vogel**[3], **Iris Huang**[1], **Shanan N. Emmanuel**[1¤], **Jesse Plung**[1], **Lillianna Radoshevich**[3], **Jasmine T. Perez**[1], **Anice C. Lowen**[2], **Balaji Manicassamy**[3]*

**1** Department of Microbiology, University of Chicago, Chicago, Illinois, United States of America,
**2** Department of Microbiology and Immunology, Emory University School of Medicine, Atlanta, Georgia, United States of America, **3** Department of Microbiology and Immunology, University of Iowa, Iowa City, Iowa, United States of America

¤ Current address: Department of Biochemistry and Molecular Biology, University of Florida, Gainesville, Florida, United States of America
* balaji-manicassamy@uiowa.edu

**Data Availability Statement:** All relevant data are within the manuscript and its Supporting Information files.

## Abstract

It is well documented that influenza A viruses selectively package 8 distinct viral ribonucleo-protein complexes (vRNPs) into each virion; however, the role of host factors in genome assembly is not completely understood. To evaluate the significance of cellular factors in genome assembly, we generated a reporter virus carrying a tetracysteine tag in the NP gene (NP-Tc virus) and assessed the dynamics of vRNP localization with cellular components by fluorescence microscopy. At early time points, vRNP complexes were preferentially exported to the MTOC; subsequently, vRNPs associated on vesicles positive for cellular factor Rab11a and formed distinct vRNP bundles that trafficked to the plasma membrane on microtubule networks. In Rab11a deficient cells, however, vRNP bundles were smaller in the cytoplasm with less co-localization between different vRNP segments. Furthermore, Rab11a deficiency increased the production of non-infectious particles with higher RNA copy number to PFU ratios, indicative of defects in specific genome assembly. These results indicate that Rab11a+ vesicles serve as hubs for the congregation of vRNP complexes and enable specific genome assembly through vRNP:vRNP interactions, revealing the importance of Rab11a as a critical host factor for influenza A virus genome assembly.

## Author summary

The influenza A virus (IAV) genome is composed of 8 distinct RNA segments. It has remained unclear how these 8 individual RNA segments are assembled together to form infectious virus particles. Our study shows that Rab11a+ vesicles serve as platforms for the congregation and assembly of 8 individual viral RNA segments needed to form infectious

**Funding:** JH and OV were partly supported by the NIH Molecular and Cellular Biology training program at The University of Chicago (T32GM007183) and the NIH Diversity Supplement (R01AI123359- 02S1). AL is supported by NIAID grant R01AI125268 and CEIRS contract HHSN272201400004C. BM is supported by NIAID grants (R01AI123359 and R01AI127775). The funders had no role in study design, data collection and analysis, decision to publish, or preparation of the manuscript.

**Competing interests:** The authors have declared that no competing interests exist.

virus particles. However, in cells lacking Rab11a, viral RNA segments fail to congregate together, resulting in increased production of defective virus particles, likely due to misassembling of viral RNA segments. Thus, our study reveals the important role for Rab11a in influenza virus genome assembly and production of infectious virus particles.

## Introduction

The segmented nature of the influenza A virus (IAV) genome poses a challenge, as it necessitates selective and efficient packaging of 8 distinct genomic RNAs into a nascent virion to form an infectious virus particle. IAV are also atypical in that genome replication occurs in the nucleus, whereas virion packaging and budding occurs at the plasma membrane [1]. The current model for IAV genome packaging suggests that newly synthesized viral ribonucleoprotein complexes (vRNPs) are exported out of the nucleus and trafficked across the cytoplasm to the plasma membrane for egress [2,3]. vRNPs are trafficked on vesicles that are positive for the host factor Rab11a using the microtubule or actin networks [4–7]. It has been proposed that association of the 8 individual vRNPs occurs during transport across the cytoplasm; however, studies also suggest that the process of vRNP association can begin in the nucleus [6–9]. Thus, it remains to be determined how 8 distinct vRNP complexes are brought together for specific genome assembly.

Recombinant IAV carrying fluorescent reporters have been instrumental in our understanding of virus-host interactions. Previously, we and others have developed reporter viruses with fluorescent tags on the non-structural protein 1 (NS1), polymerase acidic protein (PA), and polymerase basic protein 2 (PB2), and have utilized these reporter viruses to reveal key features of IAV pathogenesis and vRNP trafficking [7,10–12]. To enhance our ability to visualize vRNPs during trafficking, nucleoprotein (NP) could serve as an alternate candidate for a fluorescent tag, as numerous NP molecules bind to viral genomic RNA to form vRNPs. Unfortunately, there are no reports describing the fusion of a fluorescent tag to NP, as NP is less amenable to the addition of foreign sequences [13]. Thus, developing new strategies to fluorescently tag NP will allow us to investigate how cellular factors interact with vRNP complexes to facilitate the transport, assembly, and packaging of vRNPs into nascent virions.

In this study, we generated an influenza reporter virus with a tetracysteine (Tc; CCPGCC) tag in the NP gene (NP-Tc) and analyzed the dynamics of vRNP trafficking by utilizing FlAsH (fluorescence arsenical hairpin binder), a small molecule dye that becomes intensely fluorescent upon binding to the Tc tag. This Tc-FlAsH system allows for the visualization of a protein of interest through the addition of a short peptide tag that would minimally impact protein function [14]. We utilized this new tool to investigate how distinct vRNP complexes come together for specific genome assembly. FlAsH staining of NP-Tc virus infected A549 (human lung epithelial) cells revealed distinct puncta in the cytoplasm that were positive for viral genomic RNA, demonstrating that the NP-Tc virus can be used to visualize vRNP complexes. A time course analysis of NP-Tc virus infected cells showed that vRNPs exported from the nucleus, clustered around the microtubule-organizing center (MTOC), then trafficked on Rab11a+ vesicles via microtubule networks in an organized manner. Importantly, these data support the model previously put forth that Rab11a+ vesicles serve as cytoplasmic hubs for the bundling of vRNPs, thus promoting interactions between individual vRNP segments [5,7,15,16]. As such, vRNP bundles were smaller and dispersed in the cytoplasm of Rab11a CRISPR/Cas9 KO cells, with no significant co-localization between different vRNP segments. Analysis of released virus particles revealed that the RNA copy number to PFU ratios were

significantly higher in viruses derived from Rab11a KO cells as compared to those from WT cells, indicating that in the absence of Rab11a misassembled viral genomes are incorporated into nascent particles. Together, these data suggest that the host factor Rab11a is crucial for specific assembly of IAV genomes.

## Results

### Design and characterization of tetracysteine-tagged NP reporter virus

IAV NP is important for transcription and replication of viral RNA segments. To determine if NP can be tagged with foreign sequences without significantly affecting function, we C-terminally tagged NP with a small 6-amino acid tetracysteine tag CCPGCC (NP-Tc) or with RFP (NP-RFP) and observed expression levels like wild-type NP in transfected cell lysates (Fig 1A). Next, we evaluated the ability of NP to support replication of a reporter construct mimicking vRNA in a mini-genome assay. The NP-Tc construct supported nearly 70% reporter activity in the mini-genome assay as compared to wild-type NP (Fig 1B); however, NP-RFP showed very little reporter activity in the mini-genome assay, suggesting that the addition of a larger C-terminal tag such as RFP impaired NP function. Next, to generate a recombinant A/Puerto Rico/8/1934 (PR8) virus carrying Tc-tagged NP, we engineered an NP-Tc segment in which the small Tc tag was introduced right before the Stop codon (Fig 1C). As insertion of the Tc-tag may disrupt the 5' NP vRNA packaging sequence, we duplicated the 200nt sequence at the 5' end (negative sense genomic RNA) of the NP segment to maintain the necessary packaging signals [17]. As a control, we generated an NP segment with a duplicated packaging sequence but no TC tag (NP200, Fig 1C). Next, using the IAV reverse genetics system, we rescued recombinant NP-Tc virus, purified individual plaques, and amplified virus stocks in 10-day old embryonated eggs. NP-Tc virus reached approximately $2x10^6$- $4x10^6$ PFU/ml in allantoic fluids, a titer 100-1000-fold lower than titers obtained for wild type (WT) PR8 or NP200 control viruses. We also confirmed the presence of the Tc-tag in the purified virus stocks by sequencing of viral genomic RNA. Next, we compared the growth kinetics of NP-Tc, NP200, and WT PR8 viruses in MDCK cells. In this multi-cycle replication assay, NP-Tc virus reached peak titers of $\sim10^6$ PFU/mL, as compared to peak titers of $\sim10^8$ PFU/mL for WT PR8 or NP200 virus (Fig 1D). These data indicate that, although NP-Tc virus demonstrated decreased replication, the Tc-tagged virus was capable of undergoing multicycle replication.

To assess the potential of NP-Tc virus to serve as a new tool to visualize NP dynamics during replication, we infected A549 cells with NP-Tc virus, stained with FlAsH reagent, and performed fluorescence microscopy. We observed NP-Tc in distinct green, fluorescent puncta in the cytoplasm of infected cells, which also co-stained with an NP antibody (Fig 1E). A549 cells infected with NP200 virus showed similar NP puncta in the cytoplasm, suggesting that the NP-Tc puncta represent genuine NP structures formed during infection. As it is possible for influenza reporter viruses to lose foreign genes, we performed a passaging experiment in MDCK cells to determine the stability of the Tc-tag. After 10 passages (p10) in MDCK cells, we compared the parental NP-Tc (p1) and passaged NP-Tc virus (p10) using fluorescence microscopy (Fig 1F). Both NP-Tc p1 and NP-Tc p10 showed similar patterns of staining with FlAsH and NP antibody, demonstrating that the Tc-tag was stable in NP-Tc virus over multiple passages (Fig 1E). In addition, sequencing of the NP-Tc p10 segment indicated no changes in NP-Tc coding sequence after the tenth passage. These data demonstrate that the Tc-tag accurately marked NP and remained stably integrated and functional throughout passaging. Together, these results indicate that NP-Tc virus, despite decreased peak titers, was capable of undergoing multiple rounds of replication and could be used to visualize NP complexes during infection.

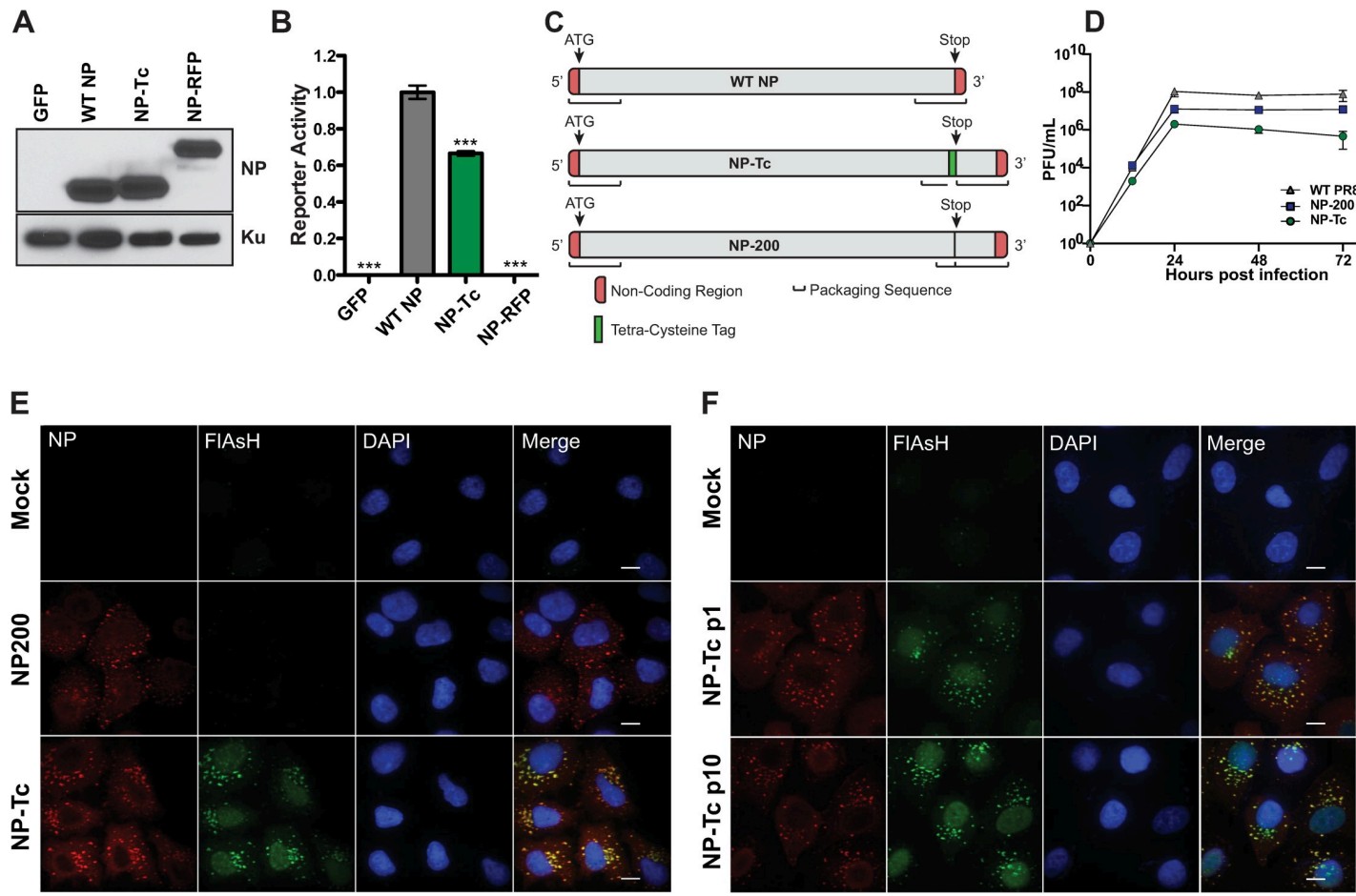

**Fig 1. Design and characterization of Tc-tagged NP virus.** (A) Western blot analysis of fluorescently tagged NP. HEK293T cells were transfected with different NP constructs and cell lysates were prepared at 48h post-transfection. Western blot analysis was performed using anti-NP antibody and Ku antigen is shown as a loading control. (B) Mini-genome activity of fluorescently tagged NP. HEK293T cells were transfected with plasmids expressing PB1, PB2, and PA of H1N1 (A/WSN/1933) along with WT NP or fluorescently tagged NP constructs and a firefly luciferase vRNA reporter. Transfection efficiency was normalized using an SV40-renilla reporter. Data are represented as mean reporter activity relative to WT NP (n = 6 per condition ± SD). (C) Schematics of modified NP segments. WT NP segment contains packaging sequences at both 5' and 3' termini. As the insertion of a Tc tag disrupts the 3' packaging sequence, the 3' packaging sequence (200 nucleotides) was duplicated after the stop codon. Schematics not to scale. (D) Comparison of growth kinetics for WT PR8, NP200 and NP-Tc viruses. MDCK cells were infected with WT PR8 or NP-Tc virus at MOI = 0.01 and viral titers were measured at the indicated time points by plaque assay. Limit of detection = 10 PFU. Data are represented as mean titers of triplicate samples ± SD. (E) NP-Tc FlAsH stain co-localizes with NP antibody staining. A549 cells were infected with NP-Tc or NP200 virus at MOI = 1 for 9 hours and stained with FlAsH dye and anti-NP antibody. Scale bar = 10 μm. (F) Tc tag remains stable in NP-Tc virus after 10 passages. NP-Tc virus was blindly passaged in MDCK cells 10 times. After passage 10 (p10), individual plaques were purified and amplified in embryonated eggs. A549 cells were infected with virus from passages 1 and 10 at MOI = 1 for 7 hours and stained with FlAsH dye and anti-NP antibody. Scale bar = 10 μm. * p-value < 0.05; ** p-value < 0.01; *** p-value < 0.001; ns, non-significant. Data are representative of at least three independent experiments.

## NP-Tc virus allows for visualization of vRNP dynamics

To determine if the cytoplasmic puncta visualized in NP-Tc virus infected cells were bona fide vRNP complexes, we performed fluorescence *in situ* hybridization (FISH) using probes against the NP vRNA segment (genome) along with FlAsH-staining. A549 cells were infected with NP-Tc virus (MOI = 1) and stained with FlAsH reagent at 7hpi followed by FISH analysis. The cytoplasmic NP-Tc puncta observed by FlAsH staining showed the presence of viral genomic RNA via FISH staining, indicating that the FlAsH-positive NP-Tc puncta were indeed vRNP complexes (Fig 2A). Next, we followed the dynamics of vRNP complex formation through a time course of infection. We observed small NP-Tc puncta outside the nucleus as early as 3hpi,

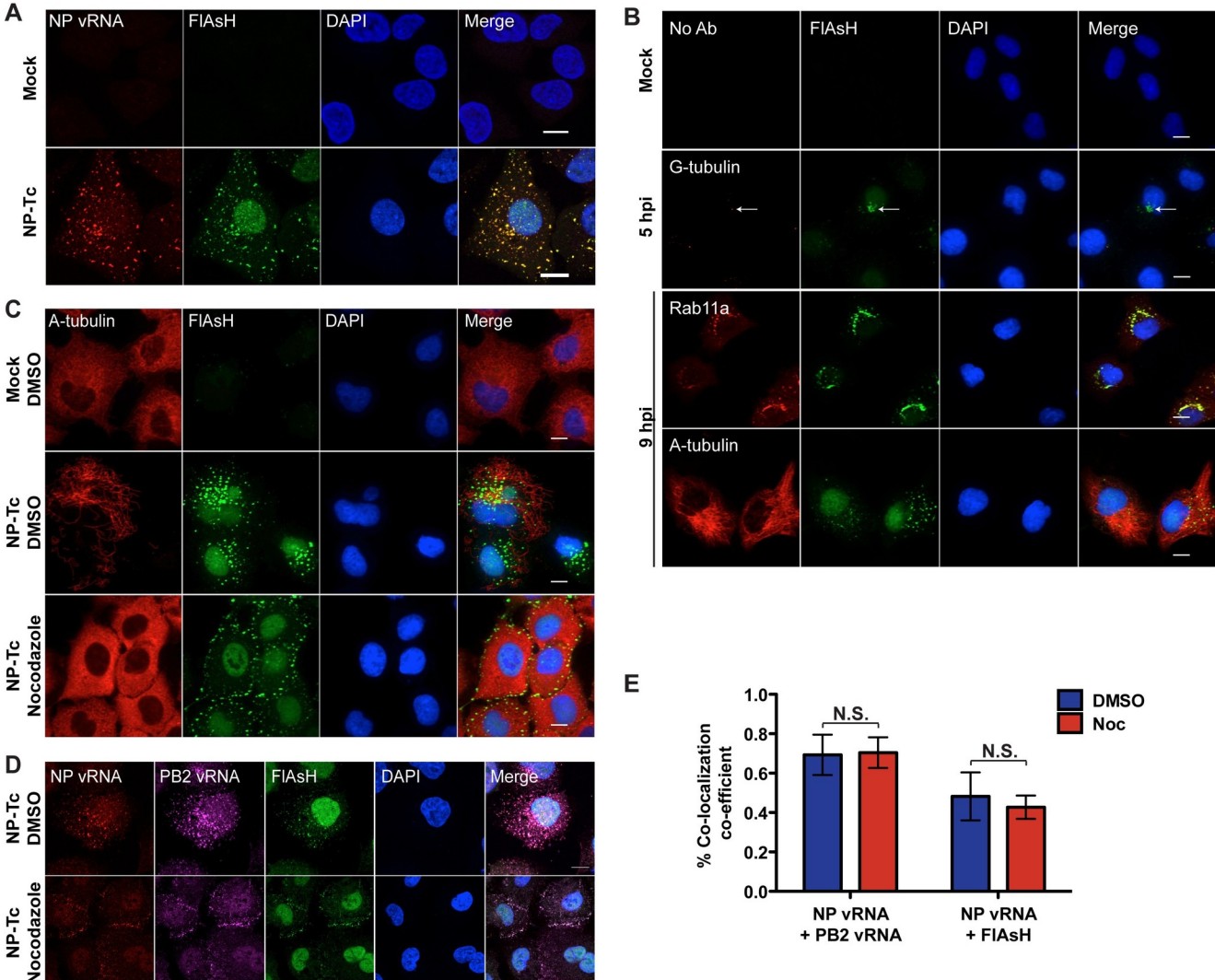

**Fig 2. NP-Tc tag allows for visualization of vRNP interactions with cellular components.** (A) FlAsH staining of NP-Tc marks vRNP complexes. A549 cells were infected with NP-Tc virus at MOI = 1 for 7 hours, and stained with FlAsH dye and FISH (Quasar570) probes against NP vRNA. Scale bar = 10 μm. (B) Localization of vRNP complexes with cytoskeletal structures and trafficking factors. A549 cells were infected with NP-Tc virus at MOI = 1 for the indicated time points, and stained with FlAsH dye, and antibodies for the cellular factors γ-tubulin, Rab11a, and α-tubulin. Scale bar = 10 μm. (C) Disruption of the microtubule network disperses vRNPs to the plasma membrane. A549 cells infected with NP-Tc virus at MOI = 1 were treated with 30μM nocodazole or DMSO starting at 3 hpi and were stained with FlAsH dye and an α-tubulin antibody at 9 hpi. Scale bar = 10 μm. (D and E) Association of vRNPs may not rely on the microtubule network. A549 cells infected with NP-Tc virus at MOI = 1 were treated with 30μM nocodazole (Noc) or DMSO starting at 3 hpi. At 8 hpi, cells were stained with FlAsH dye and FISH probes against NP vRNA (Quasar570) and PB2 vRNA (Quasar670). Scale bar = 10 μm. Cytoplasmic signals were analyzed for co-localization by fluorescence microscopy (D) and through Manders Coefficient analysis for FlAsH (NP-Tc) with PB2 or NP vRNA FISH probes (E). At least 9 cells per condition were analyzed. * p-value < 0.05; ** p-value < 0.01; *** p-value < 0.001; N.S., non-significant. Data are representative of at least three independent experiments.

indicating the initiation of vRNP export from the nucleus (S1 Fig). Over time, we observed an increase in both the sizes and numbers of NP-Tc puncta in the cytoplasm, suggesting the likely formation of supramolecular complexes of multiple vRNPs or vRNP bundles (S1C and S1D Fig). Subsequently, these large NP-Tc puncta trafficked away from the nucleus towards the plasma membrane in an organized fashion (S1 Fig). Taken together, NP-Tc virus can serve as a novel tool to assess vRNP trafficking across the cytoplasm during infection.

## Cytoskeletal components aide in the regulated transport of vRNPs

Previous research indicates that vRNPs export from the nucleus to Rab11a+ vesicles around the microtubule organizing center (MTOC) and transport along microtubules to the plasma membrane, forming vRNP sub-bundles and supramolecular structures to increase selective vRNA:vRNA interactions during transit [3,18,19]. To further validate the NP-Tc system, we examined the role of the microtubule network in vRNP trafficking by co-staining for different components of the cytoskeletal network. Using γ-tubulin as a marker for the MTOC, we observed accumulation of NP-Tc puncta near the MTOC, with NP-Tc puncta encircling γ-tubulin positive puncta at ~3–5 hpi (Fig 2B). These data confirm that vRNP complexes are exported from the nucleus to the MTOC. At later times during infection, NP-Tc puncta increased in size and co-localized with the cellular factor Rab11a prior to organized trafficking on the microtubule network (Figs 2B and S1C). As recently reported, we also observed association of ER exit site marker Sec31a in the periphery of some of the FlAsH puncta (S2 Fig) [16]. To assess this new system for live-cell imaging, we infected WT A549 cells expressing mCherry-Rab11a with NP-Tc virus and visualized vRNPs by FlAsH staining (S1 Movie). We observed association of FlAsH puncta and mCherry signal, demonstrating the association of vRNPs with Rab11a+ vesicles. In addition, we also observed coalescence of vRNP puncta as well as splitting-off of vRNP puncta, as previously reported by others (S3 Fig) [12]. As the Tc-FlAsH system is sensitive to photobleaching, we could only perform live-cell imaging of vRNP association with mCherry-Rab11a for a short time, as vRNP signal was completely faded after 30min [20].

Next, we assessed the importance of the microtubule network for vRNP trafficking by treating NP-Tc virus infected A549 cells with the microtubule depolymerizing agent nocodazole starting at 3hpi. At 9hpi, nocodazole treatment resulted in the loss of organized NP-Tc puncta structures in the cytoplasm, with the majority of large NP-Tc puncta observed at the plasma membrane, which is in agreement with previous findings (Fig 2C) [4,5,21,22]. To determine if nocodazole treatment diminished association of different vRNP complexes, we performed FISH analysis for NP and PB2 vRNA segments. We observed no significant difference in co-localization of NP/PB2 segments with and without nocodazole treatment, through Manders co-efficient analysis (Fig 2D and 2E). These data demonstrate that the microtubule network is preferred, but not absolutely required, for vRNP trafficking across the cytoplasm. Importantly, trafficking of vRNP complexes in NP-Tc virus infected cells was identical to the egress of WT IAV strains as reported by other groups (S1 Fig) [4,5,21,22].

## Rab11a knock-out A549 cells demonstrate decreased viral replication

Prior studies indicate that the Rab11a recycling pathway is important for transport of vRNP complexes to the plasma membrane [11,23–25]. The majority of the studies investigating Rab11a utilized either siRNA mediated knockdown or overexpression of Rab11a mutants [5,23,24]. To further investigate the importance of Rab11a in the NP-Tc system, we generated Rab11a KO A549 cells using the CRISPR/Cas9 technique (Fig 3A). As Rab11a is important for slow recycling of transferrin, we first performed a transferrin uptake assay as previously described [26]. We observed both lower initial binding and recycling of transferrin in Rab11a KO cells as compared to control cells, demonstrating that the slow endosomal recycling pathway is impaired in the absence of Rab11a (Fig 3B). However, we did not observe any morphological changes or growth defects in Rab11a KO cells for the 25 tested passages as compared to control A549 cells, as previously reported for other cell types [27–29]. Next, we assessed the replication of different IAV strains in Rab11a KO and control cells. In three independent Rab11a KO clones, infection with an H5N1 strain (A/Vietnam/1203/04, low pathogenic;

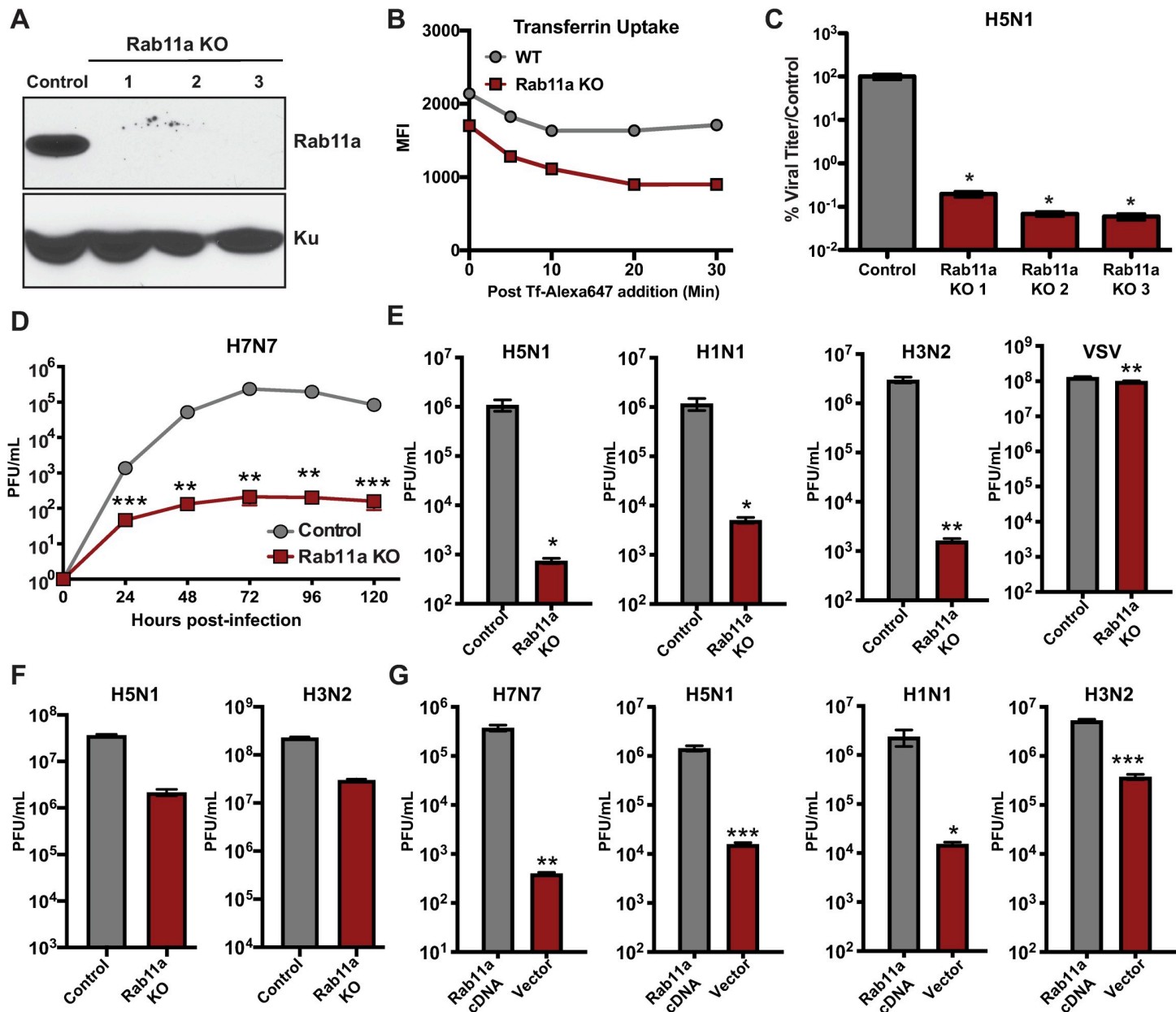

**Fig 3. Rab11a is essential for the replication of multiple IAV strains.** (A) Western blot analysis of Rab11a expression in control and Rab11a KO A549 cells. Cell lysates from control A549 cells and three independent clones of Rab11a KO cells were subjected to western blot analysis for Rab11a expression. Ku expression was used as a loading control. (B) Rab11a KO cells show reduced transferrin uptake. Serum starved control A549 and Rab11a KO cells were incubated with labelled transferrin for 20 min and transferrin levels at the indicated times were measured by flow cytometry. Values are shown as mean fluorescent intensity (MFI). (C) H5N1 replication is reduced in Rab11a KO cells. Control A549 cells and Rab11a KO cells were infected with H5N1 (MOI = 0.001) and viral titers were measured at 48 hpi by plaque assay. Viral titers are shown as mean % relative to titer from control cells ± SD. (D) H7N7 shows decreased replication kinetics in Rab11a KO cells. Control A549 and Rab11a KO cells were infected with H7N7 (MOI = 0.01) and viral titers were measured at the indicated time points. (E) Replication of different IAV strains is reduced in Rab11a KO cells. Control A549 and Rab11a KO cells were infected with H5N1 (MOI = 0.001), H1N1 (MOI = 0.01), H3N2 (MOI = 0.01), and VSV (MOI = 0.001) and viral titers were measured at 48 hpi by plaque assay. (F) Single cycle replication of different IAV strains in Rab11a KO cells. Control A549 and Rab11a KO cells were infected with H5N1 (MOI = 1), and H3N2 (MOI = 5), and viral titers were measured at 24 hpi by plaque assay. (G) Complementation of Rab11a expression in Rab11a KO cells restores IAV replication. Rab11a KO cells complemented with empty vector or Rab11a cDNA were infected with H5N1 (MOI = 0.001), H1N1 (MOI = 0.01), and H7N7 (MOI = 0.01) and viral titers were measured at 48 hpi. For panels C-E, the limit of detection = 10 PFU. Data are represented as mean titer of triplicate samples ± SD. * p-value < 0.05; ** p-value < 0.01; *** p-value < 0.001; ns, non-significant. Data are representative of at least three independent experiments.

MOI = 0.001) showed ~1000-fold lower replication as compared to control cells (Fig 3C). In addition, the replication kinetics of an H7N7 strain (A/Netherlands/219/03, low pathogenic) in Rab11a KO cells showed little increase in viral titer over time, with titers remaining at ~$10^2$ PFU/mL, whereas H7N7 replication increased steadily and reached peak titers of $10^5$ PFU/mL in control cells (Fig 3D). Moreover, replication of several additional IAV strains under low MOI conditions was also reduced by 100-1000-fold in Rab11a KO cells as compared to control cells (Fig 3E). In contrast, the replication of vesicular stomatitis virus (VSV) in Rab11a KO and control cells was similar (Fig 3E), demonstrating that Rab11a is specifically required for the replication of diverse IAV strains. Previous Rab11a siRNA knock down (KD) studies showed a 5-10-fold decrease in viral titers for high MOI infections [5,24]. To allow comparison to this prior work, we performed high MOI infections in Rab11a KO cells and similarly observed a 10-fold decrease in viral titers for H1N1 and H5N1 viruses (Fig 3F). To confirm that the reduced viral replication observed in Rab11a KO cells was specifically due to the loss of Rab11a, we ectopically expressed HA-tagged Rab11a in Rab11a KO cells and evaluated viral replication (Fig 3G). Reconstitution of Rab11a KO cells with Rab11a cDNA restored H5N1, H1N1, and H7N7 replication by >100 fold as compared to Rab11a KO cells expressing empty vector (Fig 3G), indicating that the reduced IAV replication observed in Rab11a KO cells was specifically due to the absence of Rab11a. Together, these data highlight Rab11a is a crucial host factor for efficient replication of diverse IAV strains.

## Loss of Rab11a disrupts vRNP bundling

We next sought to determine the steps of the IAV life-cycle affected by loss of Rab11a. First, we assessed the susceptibility and permissiveness of Rab11a KO cells by single cycle infection experiments using different IAV expressing GFP or luciferase reporters [10,30]. Both H1N1-GFP and H5N1-GFP viruses showed similar levels of infection (% GFP cells) in Rab11a KO and control A549 cells (S4A Fig). In addition, Rab11a KO cells infected with H1N1-Gluc virus showed similar levels of luciferase expression as compared to control A549 cells, demonstrating that IAV entry and genome replication are unaffected by loss of Rab11a (S4B Fig). As Rab11a is involved in vesicle trafficking, we assessed the cell surface expression of viral HA, NA, and M2 proteins and observed equal expression between Rab11a KO and control A549 cells (S4C Fig). To assess the functional consequences of loss of Rab11a on vRNP trafficking, we infected Rab11a KO cells with NP-Tc virus (MOI = 1), and analyzed vRNP localization by FlAsH and FISH staining at 9hpi (Fig 4A). In both WT and Rab11a KO cells, we observed intense FlAsH staining in the nucleus of NP-Tc infected cells, which is likely due to binding of FlAsH to free NP-Tc proteins in the nucleus. Interestingly, Rab11a KO cells showed either no FlAsH/FISH puncta or small, diffuse puncta in the cytoplasm, indicating that loss of Rab11a likely prevents the formation of large supramolecular vRNP complexes in the cytoplasm (Fig 4A). In addition, faint, cytoplasmic staining of NP and PB2 FISH probes did not appear to co-localize, demonstrating the importance of Rab11a for association of different vRNP segments. Moreover, Rab11a KO cells did not show significant accumulation of large FlAsH puncta in the cytoplasm at 16hpi (S4D Fig). In contrast, control cells infected with NP-Tc virus showed significant accumulation of supramolecular vRNP structures in the cytoplasm in an organized pattern (Figs 4A and S4D). To further confirm these findings, we infected Rab11a KO and control A549 cells with WT PR8 and visualized vRNPs by FISH staining. We observed significantly smaller vRNP puncta in Rab11a KO cells as compared to control cells (Fig 4B and 4C). These data suggest that Rab11a+ vesicles may act as cytoplasmic hubs for vRNP assembly.

To test whether the lack of large vRNP bundle formation in Rab11a KO cells was the result of decreased vRNP levels in the cytoplasm, we infected cells with NP-Tc virus and performed

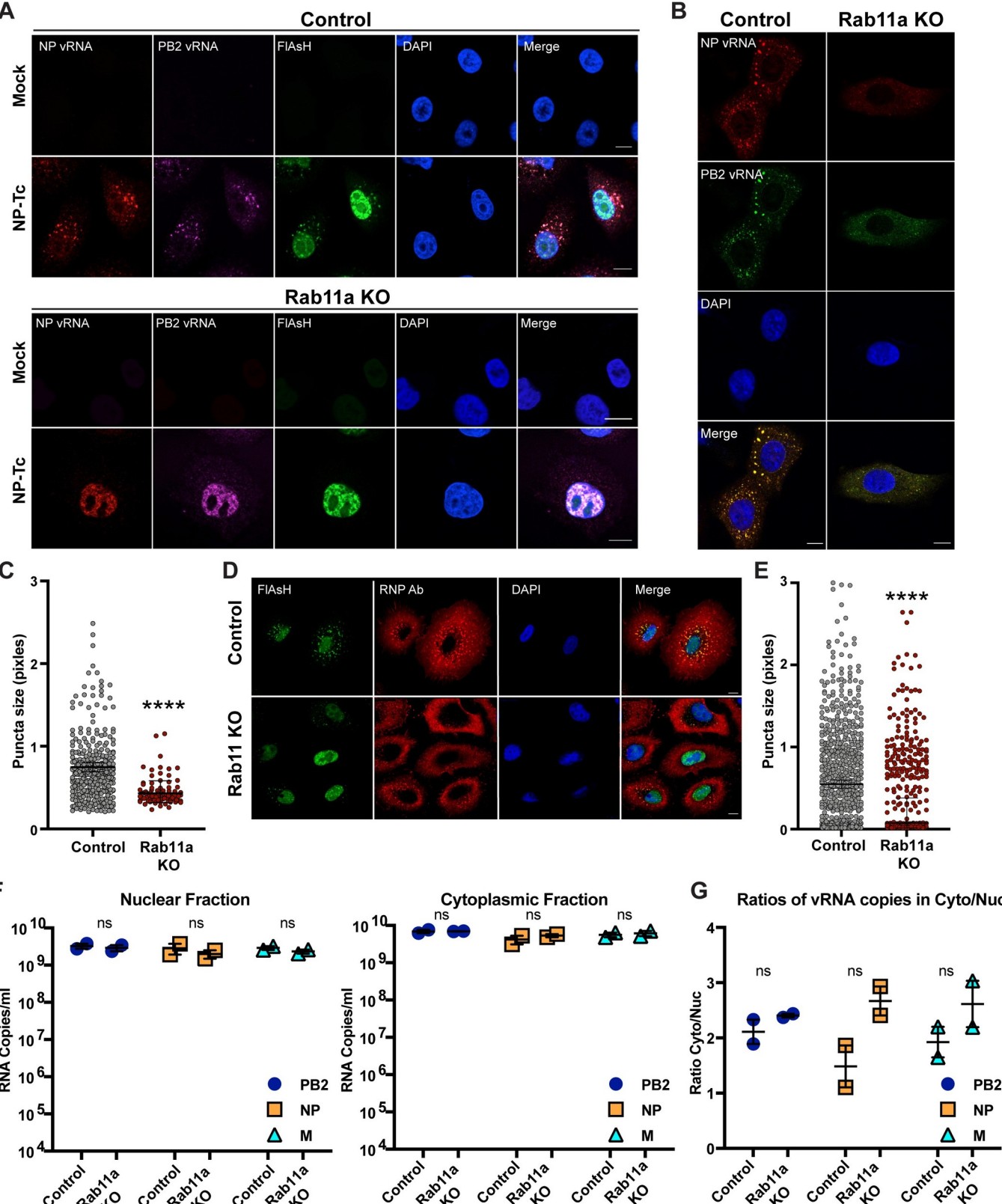

**Fig 4. Absence of Rab11a affects IAV genome assembly.** (A) vRNP complexes are scattered in the cytoplasm of Rab11a KO cells. Control A549 and Rab11a KO cells were infected with NP-Tc virus at MOI = 1 for 9 hours. Cells were stained with FlAsH dye to mark NP-Tc as well as with FISH probes against NP vRNA

(Quasar570) and PB2 vRNA (Quasar670). Scale bar = 10 μm. (B) WT PR8 forms small vRNP puncta in Rab11a KO cells. Control A549 and Rab11a KO cells were infected with WT PR8 at MOI = 10 for 9 hours and cells were stained with FISH probes against NP vRNA (Quasar570) and PB2 vRNA (Quasar670). (C) Sizes of vRNP puncta are smaller in Rab11a KO cells. The sizes of individual puncta that showed co-localization of NP and PB2 vRNA FISH probes were measured in pixels. A total of 20 cells were analyzed for each cell type. (D) RNP antibody staining confirms small and diffused RNP distribution in the cytoplasm of Rab11a KO cells. Infections were performed as described in panel A and at 16hpi, cells were stained with FlAsH dye and an RNP specific antibody. Scale bar = 10 μM. (E) Sizes of vRNP puncta are smaller in Rab11a KO cells. The sizes of individual puncta stained by the RNP specific antibody were measured in pixels. A total of 20 cells were analyzed for each cell type. (F) vRNA levels in the nuclear and cytoplasmic fractions are unaffected in Rab11a KO cells. Control A549 or Rab11a KO cells were infected with WT PR8 at MOI = 5. At 7hpi, RNA from the cytoplasmic and nuclear fractions was isolated and copy numbers for PB2, NP, and M segments were determined by digital droplet PCR. (G) Ratios of vRNA copy numbers in the cytoplasmic and nuclear fractions. * p-value < 0.05; ** p-value < 0.01; *** p-value < 0.001; N.S., non-significant. Data are representative of at least two independent experiments.

FlAsH and RNP specific antibody staining. As before, we did not detect large FlAsH puncta in the cytoplasm of Rab11a KO cells; however, we readily detected RNPs in the cytoplasm via staining with RNP specific antibodies (Fig 4D). This is likely due to the decreased sensitivity of FlAsH staining as compared to antibody staining. RNP puncta in Rab11a KO cells appeared smaller and/or diffused by antibody staining as compared to control A549 cells, demonstrating cytoplasmic dispersion of RNPs in the absence of Rab11a (Fig 4D and 4E). To further demonstrate that Rab11a KO cells do not have any defects in the export of vRNPs to the cytoplasm, we isolated the nuclear and cytoplasmic fractions from WT PR8 infected Rab11a KO cells and control A549 cells and determined the copy numbers of vRNPs by digital droplet PCR (7hpi; MOI = 5). We did not observe significant differences in the copy numbers of PB2, NP, and M vRNA segments in both nuclear and cytoplasmic fractions between Rab11a KO and control A549 cells (Fig 4F). In addition, the ratios of vRNA copy numbers in the cytoplasmic and nuclear fractions were similar, demonstrating that Rab11a KO cells do not have any defect in exporting vRNPs to the cytoplasm (Fig 4G). The relative enrichment for nuclear and cytoplasmic fractions in mock and WT PR8 infected cells were confirmed by semiquantitative RT-PCR for U6 and S14 RNAs, respectively (S5 Fig). Together, these results demonstrate that vRNPs fail to from supramolecular vRNP bundles in the absence of Rab11a, despite Rab11a KO cells showing similar vRNP levels as control cells in the cytoplasm.

## Loss of Rab11a results in defective assembly of vRNP segments and increases production of non-infectious particles

As Rab11a KO cells showed defects in vRNP bundling, we hypothesized that, in the absence of Rab11a, individual vRNP segments are scattered across the cytoplasm and are unable to efficiently associate in a specific manner, ultimately resulting in the production of defective particles with mis-incorporated vRNPs. To test this hypothesis, we assessed the levels of non-infectious particles in the supernatant from Rab11a KO and control A549 cells by measuring the ratios of infectious particles via plaque assay to total particles via hemagglutination assay (Fig 5A). Supernatants from WT PR8 infected Rab11a KO cells showed an ~20-fold lower ratio of PFU to HA units (HAU) as compared to supernatants from infected control A549 cells, indicating that there were higher numbers of non-infectious particles in the supernatants obtained from infected Rab11a KO cells (Fig 5A). In addition, we assessed the relative levels of defective particles in the supernatants by measuring the ratio of genome copy number to infectious particles (PFU). We observed 2-3-fold higher vRNA copy number to PFU ratios for viral supernatants from Rab11a KO cells as compared to control A549 cells for both H1N1 and H3N2 (S6A and S6B Fig). Furthermore, to remove potential contaminating viral genome released from dead cells that could contribute to the observed differences, we purified the virions on a 25% sucrose cushion and measured the ratios of vRNA copy number to PFU or HAU. Rab11a KO and control cells were infected with H1N1 (PR8) at an MOI = 0.01 and at 72hpi, the virions in the supernatants were pelleted on a 25% sucrose cushion. As anticipated,

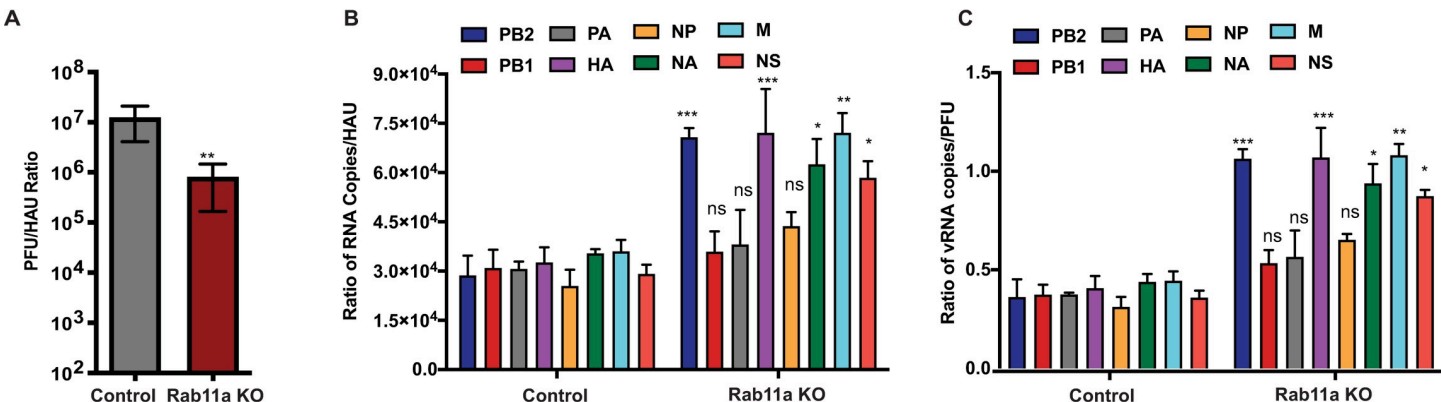

**Fig 5. Absence of Rab11a results in defective genome assembly and increased production of defective particles.** (A) Rab11a KO cells show increased production of non-infectious virus particles. Control A549 and Rab11a KO cells were infected with WT PR8 at MOI = 0.01 and viral titers in the supernatants were measured at 48 hpi by plaque assay. Hemagglutination titers were determined as HA units (HAU) using 0.5% chicken RBC. The ratio of PFU to HAU is shown. (B-C) Virions purified from Rab11a KO cells show increased defective particles. Control A549 and Rab11a KO cells were infected with WT PR8 (H1N1) for 72 hours, and virions in the supernatants were purified on a sucrose cushion. Viral genomic RNAs were reverse transcribed and individual segment copy numbers were quantified by digital droplet PCR. (B) Ratio of vRNA copy numbers to HAU. (C) Ratio of vRNA copy numbers to PFU. Ratios are shown for all eight PR8 segments. * p-value < 0.05; ** p-value < 0.01; *** p-value < 0.001; N.S., non-significant. Data are representative of three replicates of virion preparations from control A549 and Rab11a KO cells.

we obtained lower levels of infectious particles from Rab11a KO cells, which resulted in decreased genome copies (S6C Fig); the eight individual vRNA segments were equally represented in the virions purified from Rab11a KO supernatants, indicating that there were no defects in the incorporation of specific vRNP segments into nascent particles (S6D Fig). As before, virions purified from Rab11a KO supernatants showed increased ratios of vRNA copy number to HAU or PFU as compared to virions purified from control A549 supernatants (Fig 5B and 5C). These results demonstrate that Rab11a KO cells produced higher amounts of non-infectious particles likely due to inefficient or defective genome assembly.

Taken together, we have developed a virus containing Tc-tagged NP that serves as a unique tool to selectively visualize NP in vRNP complexes and to study vRNP dynamics during infection. Importantly, our studies reveal a novel role for Rab11a in the life cycle of IAV prior to vRNP trafficking. Specifically, Rab11a+ vesicles provide a platform for vRNP bundling or genome assembly and thus facilitate efficient production of infectious IAV particles.

## Discussion

The segmented nature of the IAV genome poses a challenge, as selective and efficient packaging of eight distinct vRNP complexes is required for production of infectious particles. While several studies demonstrate that the majority of IAV particles specifically incorporate eight individual segments, host processes that promote selective viral genome assembly remain unknown [15,31–34]. Here, we have developed a novel NP-Tc reporter virus that revealed the dynamic interactions of vRNP complexes with cellular factors and cytoskeletal networks. We showed that exported vRNP complexes accumulated near the MTOC and subsequently associated with Rab11a+ vesicles to traffic to the plasma membrane via the microtubule network. Importantly, using cells lacking Rab11a, we demonstrated that Rab11a+ vesicles, which have been previously shown to aid in the trafficking of vRNP complexes across the cytoplasm, also serve as hubs for vRNP complex accumulation (Fig 6). This localized increase in vRNP concentration on Rab11a hubs likely facilitates interactions between individual vRNP segments (vRNP bundling) and promotes selective assembly of the IAV genome.

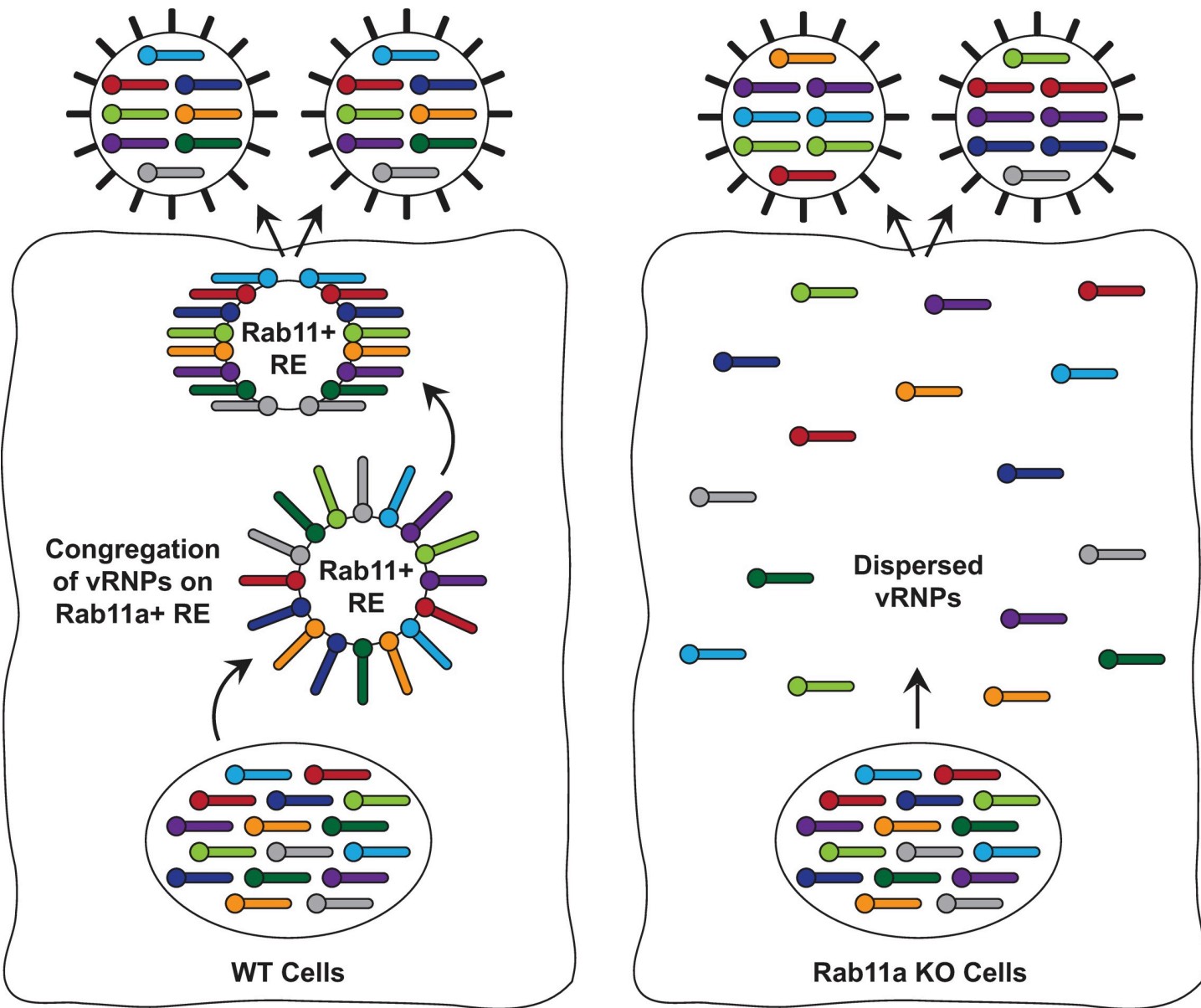

**Fig 6. Proposed model for vRNP assembly on Rab11a+ vesicles.** vRNPs exported from the nucleus congregate on Rab11a+ recycling endosomes. This increased concentration of vRNPs on Rab11a+ vesicles likely facilitates interactions between individual vRNP segments, promoting selective packaging of the viral genome. In the absence of Rab11a, vRNPs are scattered in the cytoplasm, decreasing the likelihood of association between the eight individual RNP segments, and ultimately resulting in the production of defective particles with misassembled genomes. Individual vRNPs are shown in different colors.

Fluorescently tagged viruses are useful tools to understand the intracellular biology of the virus lifecycle. Recombinant IAV carrying fluorescent tags on different viral proteins have been developed by us and others [7,10–12]. Here, we have engineered a tetra-cysteine tag into the NP gene of IAV, which allowed for the visualization of vRNP dynamics with cellular factors and cytoskeletal networks. To our knowledge, this is the first description of a recombinant IAV carrying a tagged NP gene. NP-Tc supported polymerase activity in the mini-genome assay, albeit at a 30% lower level as compared to WT NP (Fig 1B). While fusion of a small 6 amino-acid Tc tag to NP had a modest effect on NP function, addition of a larger fluorescent

protein (RFP) to NP completely abrogated NP function in the mini-genome assay (Fig 1B). As such, prior studies utilizing GFP-NP in mini-genome assays required addition of WT NP for viral polymerase activity [16]. Although NP-Tc virus replicated robustly in MDCK cells, peak viral titers for NP-Tc virus were 2-logs lower than WT PR8 or NP200 virus (Fig 1D). The modest decrease in NP-Tc polymerase function likely had a cumulative effect on viral replication, perhaps due to reduced expression of viral polymerase components (Fig 1B and 1D). Despite the decrease in NP-Tc virus replication, we were able to clearly define the interactions between vRNPs and cellular trafficking networks during IAV infection (Fig 2). Live-imaging studies with NP-Tc virus in mCherry-Rab11a expressing A549 cells showed association of FlAsH/RNP puncta with Rab11a and highlighted the dynamic nature of vRNP trafficking, which is in agreement with prior live-imaging studies performed with a PA-GFP virus (S1 Movie and S3 Fig) [12]. We observed accumulation and formation of large FlAsH/RNP puncta outside the nucleus; in addition, we observed coalescence of smaller FlAsH/RNP puncta as well as splitting-off of RNP puncta. However, due to rapid photobleaching of FlAsH dye, we could only perform live-imaging studies for 20min. Our fixed cell imaging studies showed that vRNPs appeared near the MTOC structure as early as 3–5 hpi, suggesting that vRNPs preferentially exported to the MOTC (Fig 2B). vRNPs continued to accumulate and formed large supramolecular structures that co-localized with Rab11a as well as microtubule networks (Figs 2 and S1). Subsequently, the vast majority of vRNP puncta showed directional trafficking across the cytoplasm on microtubule networks away from the nucleus. In polarized cells, Rab11a is an important mediator of apical transport on microtubule networks that facilitates the preferential assembly and release of IAV particles from apical surfaces [5,24,35]. Interestingly, disruption of the microtubule network by nocodazole treatment dispersed vRNP puncta, with the majority of vRNPs localized close to the plasma membrane (Fig 2C and 2D). These data validate previous findings that vRNPs can be trafficked to the plasma membrane independent of microtubule networks without significantly affecting co-localization of different vRNPs or attenuating viral titers [4,5,21,22]. Taken together, these studies demonstrate that the NP-Tc virus can be a valuable technology to study influenza virus genome assembly.

Several studies have shown that the Rab11a recycling endosome pathway is important for the transport of vRNP complexes to the plasma membrane [3,18,19]. Using complementary techniques including fluorescence microscopy, FRET, and immunoprecipitation, Rab11a has been shown to interact with vRNPs, and the GTPase activity of Rab11a was necessary for vRNP trafficking to the plasma membrane. The vast majority of these studies utilized siRNA KD or overexpression of a GTPase mutant to demonstrate the importance of Rab11a in vRNP trafficking [23,24,35]. siRNA mediated KD of Rab11a or over expression of mutant Rab11a has been shown to decrease viral titers by 20-fold at 16–24 hpi under high MOI conditions (MOI = 5) [24,35]. Our studies in Rab11a KO A549 cells showed a similar 10-fold decrease in viral titers during high MOI infections at 24hpi (H5N1 MOI = 1; H3N2 MOI = 5; Fig 3F), suggesting that IAV genome assembly and cytoplasmic trafficking can occur in the complete absence of Rab11a, albeit less efficiently. Interestingly, in low MOI experiments (MOI = 0.001–0.01), we observed an ~3–4 log decrease in viral titers in Rab11a KO cells as compared to control A549 cells for multiple IAV strains, demonstrating that loss of Rab11a had a cumulative effect on IAV replication (Fig 3D and 3E). This severe impairment of viral replication in Rab11a KO cells was specific to IAV strains, as replication of VSV was unaffected. Moreover, we did not observe any growth defects or morphological changes in Rab11a KO A549 cells over 25 passages as previously reported for other cell types [27–29]. By systematic analyses of different steps of the IAV lifecycle, we determined that the defects in Rab11a KO cells occur after nuclear export of vRNPs. Single cycle-infection experiments with GFP and luciferase reporter viruses demonstrated that Rab11a KO cells were susceptible and

permissive to IAV infection at levels like control A549 cells (S4 Fig); in addition, we can exclude a role for Rab11a in the expression of viral surface proteins including HA, NA, and M2, as they were unaffected in Rab11a KO cells. Likewise, we can exclude defects in vRNA production and nuclear export, as vRNA copy numbers for PB2, NP, and M segments were similar in the cytoplasmic and nuclear fractions isolated from Rab11a KO and control A549 cells (Fig 4F and 4G). Taken together, these results indicate that Rab11a may play an additional role in the IAV lifecycle, apart from vRNP trafficking, at a post nuclear export step.

Prior studies have reported co-localization of multiple vRNP segments with Rab11a, indicating that Rab11a vesicles serve as "hot spots" for vRNP accumulation [5–7,16]. In addition, vRNPs have been shown to form "viral inclusions" or "liquid organelle" like structures in the cytoplasm of infected cells in a Rab11a dependent manner [16]. Thus, it been proposed that specific assembly and packaging of eight distinct vRNA segments into nascent virions occurs through collision of different vRNP containing Rab11a+ vesicles or the dynamic exchange of materials between viral inclusions [3,7,16]. As such, upon siRNA KD or overexpression of a Rab11a GTPase mutant, vRNPs were localized to the perinuclear region or dispersed in the cytoplasm of infected cells, resulting in the decreased production of infectious virus in the supernatants [23,24,35]. These observations led to a proposed model in which Rab11a dependent vRNP "hot spots" or inclusions serve as sites for genome bundling or assembly. A recent study also demonstrated that these viral inclusions or hot spots can be formed by overexpression of a single vRNP segment in the absence of the other seven segments, suggesting that the formation of these inclusions occurs independently of vRNA-vRNA interactions between segments [16]. However, the functional importance of these hot spots or viral inclusions to viral genome assembly and infectious virion production has yet to be demonstrated. In our studies with Rab11a KO cells, vRNPs failed to form large vRNP hot spots or inclusions, and the sizes of vRNP puncta were smaller in the cytoplasm, demonstrating that Rab11a+ vesicles serve as hubs for the congregation of vRNPs (Fig 4B and 4E). As we observed smaller or dispersed vRNP puncta as well as a 3–4 log decrease in viral titers in Rab11a KO cells, we hypothesized that Rab11a is critical for the specific assembly of the IAV genome, and thus, cells lacking Rab11a will demonstrate increased production of defective particles. In agreement, failure to form large vRNP bundles increased the production of defective virions from Rab11a KO cells, as evidenced by increased ratios of vRNA copy number to PFU and decreased ratio of PFU to HAU (Figs 5 and S6). Increased production of defective particles coincided with dysregulation of vRNP stoichiometry in virions, suggesting that Rab11a promotes selective assembly of the eight viral gene segments. Results of high MOI infection experiments, in which we only observed a 10-fold decrease in viral titers, suggest that vRNPs can be incorporated into nascent virions in the absence of Rab11a (Fig 3F). Therefore, our studies demonstrate that Rab11a is an important host factor critical for efficient and selective assembly of the IAV genome.

Several seminal studies have demonstrated that IAV genome assembly occurs in a selective manner, with the genome arranged in a 7+1 configuration within the virion [31,32,36]. This selective assembly of the viral genome is proposed to occur in a hierarchical manner, likely through vRNA-vRNA contacts between segments, a finding supported by *in vitro* biochemical studies [32,36]; however, it is not completely understood where and when selective interactions between vRNPs occur during the virus life-cycle. Based on studies by us and others, we propose that Rab11a+ vesicles serve as hubs or hot spots for the congregation of vRNP complexes, and this increased concentration of vRNPs allows for selective vRNA:vRNA interactions, thereby promoting efficient and selective assembly of the IAV genome (Fig 6) [9,37]. In contrast, in the absence of Rab11a, vRNPs are scattered in the cytoplasm, decreasing the likelihood of associations occurring between the eight distinct vRNPs to facilitate selective genome

assembly. As such, this results in disorganized genome packaging into virions and increased production of defective particles (Fig 6).

In conclusion, we describe the development of a novel fluorescent reporter NP-Tc virus to study the intracellular dynamics of vRNP trafficking. We show that vRNPs preferentially associate with microtubule networks to traverse the cytoplasm on Rab11a+ vesicles. Importantly, we demonstrate that Rab11a+ vesicles serve as hubs for the congregation of vRNPs and promote selective genome assembly, as the loss of Rab11a results in increased production of non-infectious particles.

## Materials and methods

### Ethics statement

All studies were performed in accordance with the principles described by the Animal Welfare Act and the National Institutes of Health guidelines for the care and use of laboratory animals in biomedical research. The protocol for isolating RNP antibodies from mice was reviewed and approved by the Institutional Animal Care and Use Committee at the University of Iowa (Animal Protocol 8052127).

### Cell culture and viruses

Human lung epithelial cells (A549), human embryonic kidney cells (HEK293T), and African green monkey kidney epithelial cells (Vero) were cultured in 10% fetal bovine serum (FBS) and 1% penicillin/streptomycin in Dulbecco's Modified Eagle Medium (DMEM). Madin Darby canine kidney cells (MDCK) were culture in modified eagle medium (MEM) with 10% FBS and 1% penicillin/streptomycin. Various Influenza virus strains, including A/Puerto Rico/ 8/1934 (PR8, H1N1), low pathogenic version of A/Vietnam/1203/2004 (VN04, H5N1), and A/ Hong Kong/1/1968 (HK68, H3N2), were kindly provided by Dr. Adolfo Garcia-Sastre at the Icahn School of Medicine at Mount Sinai, NY. H1N1 (PB2-Gluc) PR8, H1N1-GFP (PR8) and H5N1-GFP reporter viruses were generated using the reverse genetics system [38]. Influenza A/Panama/2007/1999 virus (Pan99, H3N2) was rescued using reverse genetics. Influenza virus stocks were amplified in 10-day old specific pathogen-free eggs (Charles River) and titered in MDCK cells by standard plaque assay with 0.6% Oxoid agar or 2.4% Avicel RC-581 (provided by FMC BioPolymer, Philadelphia, PA) as previously described [39]. Influenza A/Netherlands/219/2003 virus (H7N7, low pathogenic) was rescued using reverse genetics plasmids kindly provided by Dr. Ron Fouchier at the Erasmus Medical Center in Rotterdam, The Netherlands. Dr. Glenn Barber at the University of Miami, FL kindly provided vesicular stomatitis virus expressing GFP (VSV) [40]. VSV was grown and titered in Vero cells by plaque assay with 1% methylcellulose (Sigma).

### Generation of NP-Tc and NP200 viruses

To generate the tetracysteine tag (Tc) in the NP segment, a flexible linker (GTGSGIR) followed by the 6 amino-acid Tc tag (CCPGCC) was inserted before the stop codon of the NP open-reading frame (GTGSGIRCCPGCC Stop). To maintain the NP vRNA packaging signal, the 200nt packaging sequence was duplicated [17]. The NP-Tc fragment and the 200nt packaging fragment were PCR amplified individually using the following primers: Ambi A (AAAGAT CGCTCTTCTGGGAGCAAAAGCAGG) and reverse (CAGGACAACACCGAATTCCGGA TCCGGTACCGCTAGCCCCATTGTCGTACTCCTCTGC), forward (ACCGGATCCGGA ATTCGGTGTTGTCCTGGATGTTGTTG AGTCGACGAGGACCGAAATC) and NC-Sap (AAACATCGCTCTTCTATTAGTAGAA ACAAGG). PCR products were gel extracted and

used as templates for a second round of PCR with Ambi A and NC-Sap primers. Full length PCR product was cloned by restriction digestion into pDZ vector using Sap I (Lgu I, Thermoscientific). The NP-RFP segment was generated in a similar manner; the NP, RFP, and 200nt packaging fragments were PCR amplified using the following primer pairs: Ambi A (AAAGA TCGCTCTTCTGGGAGCAAAAGCAGG) and reverse (CCGAATTCCGGATCCGGTAC CGCTAGCCCCATTGTCGTACTCCTCTGCATTG), forward (CCGGAATTCGGATGGAC GAGGATGGTTCAGAGG) and reverse (CGCGTCTAGACTATCCTCGTCGCTACCGAT GGCGC) forward (TGATCTAGAGAG GACCGAAATCATAAGG) and NC-Sap (AAACAT CGCTCTTCTATTAGTAGAA ACAAGG), respectively. The fragments were digested with restriction enzymes and cloned into pDZ using Sap I (Lgu I, Thermoscientific). Generation of the NP200 segment has been previously described [17]. Recombinant NP-Tc and NP200 viruses in the PR8 background were generated using a pDZ vector based reverse genetics system [38]. NP-Tc virus stocks were generated by amplification of plaque purified virus in eggs and the presence of the desired Tc-tag in NP was confirmed by Sanger sequencing. Briefly, viral RNA from egg grown stock or cell supernatants was isolated using the QIAamp Viral RNA Mini Kit (Qiagen) and cDNA was generated using SuperScript II Reverse Transcriptase (Invitrogen) using Ambi A primer (AAAGATCGCTCTTCTG GGAGCAAAAGCAGG). The NP segment was amplified as two fragments with the following primer pairs containing M13 sequences: NP1 forward (GTAAAACGACGGCCAGTAGCAAAAGCAGGGTAGATAATCA CTCACTG) and reverse (CACACAGGAAACAGCTATGACCATCTCTCAATATGAGTGC AGACC), NP2 forward (GTAAAACGACGGCCAGTGTGAGAATGGACGAAAAACAAG) and reverse (CACACAGGAAACAGCTATGACCATAGTAGAAACAAGGGTATTTTTCTT-TAATT). RT-PCR products were separated on a 1% agarose gel, extracted and sequenced using M13F and M13R primers.

## Mini-Genome assay

The mini-genome assay was performed by transfecting 0.5 μg each of pDZ WT NP or NP-Tc with pCAGGS PB1, PB2, and PA of A/WSN/1933 (WSN, H1N1), pPol-I firefly Luc (NP gene; 100ng) and SV40 Renilla (50ng) reporter into HEK293T cells using polyethylenimine reagent (PEI, MP Biomedicals) at a ratio of 1:5 (DNA:PEI). At 48h post- transfection, cells were lysed with 1x lysis buffer (Promega) and luciferase activity in the supernatant was measured using the Dual Luciferase Assay Kit (Promega). Firefly luciferase values were normalized for transfection efficiency to renilla luciferase values. Data are represented as firefly luciferase activity relative to WT NP ± SD. Data are representative of at least three independent experiments.

## Generation of CRISPR KO cells

Rab11a KO A549 cells were generated using two guide RNAs (gRNA) targeting the promoter and exon 1 of the Rab11a gene as previously described [39]. Oligonucleotides for the CRISPR target sites T1 (forward CACCGCATTTCGAGTAAATCGAGAC and reverse AAACGTCTC GATTTACTCGAAATGC) and T2 (forward CACCGTAACATCAGCGTAAGTCTCA and reverse AAACTGAGACTTACGCTGATGTTAC) were annealed and cloned into lenti-CRISPRv2 (Addgene #52961) and LRG (Addgene #65656) expression vectors, respectively. A549 cells transduced with lentivirus vectors expressing gRNAs were selected in the presence of 2 μg/mL puromycin for 10 days and clonal Rab11a KO cells were generated by limiting dilution of the polyclonal population. Rab11a KO cells were identified by PCR analysis of the targeted genomic region using the following primers (forward TGTTCAACCCCCTACCCCCAT TC and reverse TGGAAGCAAACACCAGGAAGAACTC) and further confirmed by western blot analysis of Rab11a expression.

## Measurement of virus replication

A549 cells were seeded in triplicate at a density of $3.5 \times 10^5$ cells/well in 12-well dishes a day prior to infection. Cells were infected at the indicated MOIs as determined by cell count prior to infection. Cells were infected in DMEM media containing 1 µg/mL TPCK-treated trypsin (T1426, Sigma), 0.01% FBS, and 0.2% bovine serum albumin (BSA). After 1 hr incubation at 37˚C, cells were washed twice in PBS and placed in DMEM/BSA media. Supernatants were collected at 48hr post-infection (hpi), serially diluted, and titered by plaque assay in MDCK cells. Infection experiments were performed with biological triplicates, data are presented as mean titer of the triplicate samples ± SD, and are representative of at least three independent experiments.

## FlAsH, antibody, and FISH staining and microscopy preparation

FlAsH staining: A549 cells were seeded onto coverslips at $1 \times 10^5$ cells/well in 24-well dishes a day prior to infection and infections were performed as described above in Opti-Modified Eagle Medium (Opti-MEM). FlAsH staining of cells was started 1hr prior to the indicated time points by incubating cells with FlAsH staining solution, consisting of 1.5 µM FlAsH (Invitrogen) and 10 µM 1,2-ethanedithiol (diluted in DMSO, Sigma), in Opti-MEM at 37˚C for 30 min. After incubation, A549 cells were washed once gently with prewarmed PBS and incubated twice with prewarmed wash buffer (500 µM 1,2-ethanedithiol diluted in DMSO and 1x 2,3-dimercapto-1-propanol (Sigma)) in Opti-MEM, each for 7 min at 37˚C. Cells were subsequently washed once with PBS and fixed with 4% Paraformaldehyde (Electron Microscopy Sciences) in PBS for 5 min at room temperature.

Antibody staining: Fixed cells on coverslips were permeabilized with 0.1% Triton X-100 (Sigma) in PBS for 15 min followed by washing in PBS and incubation in a blocking buffer consisting of 1% BSA, 0.5% fish gelatin (Sigma), and 0.01% Tween20 (Sigma) in PBS for 30 min. Permeabilized cells were incubated in primary antibodies diluted in blocking buffer for 1hr (1:1000 PR8 NP polyclonal, gift from Dr. Adolfo Garcia-Sastre; 1:1000 γ-tubulin DQ-19, #T3195 Sigma; 1:1000 α-tubulin, #T6199 Sigma; and 1:250 Rab11a, #715300 Invitrogen) followed by three washes with PBS. Subsequently, cells were incubated with secondary antibodies diluted in blocking buffer for 45 min (1:1000 Alexa-fluor 546 conjugated to mouse or rabbit, Invitrogen) followed by three PBS washes. Coverslips were mounted onto slides using ProLong Gold Antifade Mountant with DAPI (Invitrogen). All incubations involving antibody staining were performed with gentle shaking at room temperature.

Production of RNP antibodies: The RNP specific antibodies shown were made in-house by immunizing C57BL/6J mice with RNPs purified from egg gorwn PR8 virions. Briefly, PR8 virions were purified on a 25% sucrose cushion (NTE buffer 0.5 M NaCl, 10 mM Tris-HCl, 1 mM EDTA, pH 7.5) and viral membranes were disrupted with 0.1% Triton X-100 in NTE buffer. RNPs released from virions were pelleted by ultracentrifugation and resuspended in NTE buffer. C57BL/6J mice were immunized with a 1:1 mixture of RNP (5ug) and TiterMax adjuvant (Sigma) followed by a booster immunization on day 14. On day 28, serum was collected from immunized animals and high titer RNP antibody carrying animals were identified by performing immunofluorescence assay in A549 cells infected with NP-Tc virus. Mouse serum from RNP immunized animals were used in immunofluorescence assays at a 1:1000 dilution.

FISH staining: Fixed cells on coverslips were permeabilized with 0.5% Triton X-100 in 5mM MgCl$_2$-supplemented PBS (PBSM) for 1 min at room temperature. Cells were incubated in RNA wash buffer (10% formamide [Sigma] in 2xSSC [300mM sodium chloride, 30mM sodium citrate]) for 10min at room temperature. To hybridize FISH probes to cells on coverslips, 40 µL of 10 µM FISH probes in hybridization buffer (10% dextran sulfate [Sigma], 2mM

vanadyl ribonucleoside complexes [#514025, NEB], 0.02% RNAse-free BSA [NEB], 50 µg E. coli tRNA [#10706636, Roche], 10% formamide [Sigma], and 2xSSC) was applied as a droplet onto parafilm. Coverslips were inverted on top of the droplet and this complex was placed in a humidified chamber for 16 hours at 37˚C. Coverslips were washed twice, each for 30 min at 37˚C without shaking, in RNA wash buffer supplemented with 2mM vanadyl ribonucleoside complexes. Coverslips were mounted onto slides using ProLong Gold Antifade Mountant with DAPI (Invitrogen). FISH probes were generated by LGC Biosearch Technologies based on a previous publication [6]. Imaging was performed at the Integrated Light Microscopy Core Facility at the University of Chicago. The Olympus DSU Spinning Disc confocal microscope was used for general fluorescence imaging. The Leica SP8 3D 3-color Gated Super resolution confocal microscope was used for fluorescence imaging involving FISH probes.

Live cell imaging: Timelapse image series were acquired using an inverted wide-field fluorescence microscope (Axio Observer 7, Carl Zeiss Microscopy, Germany) equipped with an Axiocam 506 mono camera and the software ZEN 2.3 Pro. All experiments were carried out at 37˚C and 5% CO2 using an incubation chamber enclosing the microscope stage. 353 nm LED module was used for excitation of DAPI. 488 nm LED module was used for excitation of FlAsH dye. 548nm LED module was used for excitation of mCherry Rab11a. A series of sequential images at 14 bits were collected at a fixed pixel size of 45 nm selecting a region of interest of 900x900 pixels within the cell (i.e. corresponding approximately to 40.54x40.54 µm). We used a sampling interval of 15 s that generates in turn a frame time of 1201.05 s (N = 80 frames were acquired). For each XY position and time point, images were acquired with a with a 63x Plan Apochromat 1.40 oil immersion objective and 1.6x Tubulen optavor.

## Flow cytometric analysis

Transferrin binding assay: Control and Rab11a KO cells were serum starved for 30 min and incubated for 20 min with Alexa647 labelled transferrin (Thermo Fisher) to allow for transferrin uptake. After washing, cells were incubated for various times at 37C and the process was stopped by placing the cells on ice. Subsequently, cells were fixed with 4% formaldehyde for 10 min and transferrin levels were measured by flow cytometry using BD FACSVerse. Data was analyzed using FlowJo software (TreeStar Corp.) and shown as mean fluorescence intensity (MFI).

Analysis of GFP expression and surface levels of viral HA, NA, and M2 proteins: Control A459 or Rab11a KO cells ($1.2 \times 10^6$ cells/well) were seeded in 6-well plates a day prior and infected with H1N1-GFP (PR8 strain) or H5N1-GFP (VN04; low pathogenic) at indicated MOI for 1hr at 37C. Subsequently, cells were washed 3 times with PBS and incubated at 37C for 16hrs in DMEM/2%FBS media. The following day, cells were lifted with 0.05% trypsin and analyzed for GFP expression using BD FACSVerse. Data was analyzed using FlowJo software (TreeStar Corp.) and shown as percentage of GFP expressing cells. For surface expression of viral proteins, single cells suspensions of WT PR8 infected control A549 or Rab11a KO cells (16hrs) were incubated on ice with mouse monoclonal antibodies for 1hr (1:1,000) (anti-HA clone IC5-4F8 and anti-NA clone N2-1C1 were obtained from BEI Resources; anti-M2 clone E10 is a gift from Dr. Adolfo Garcia-Sastre) followed by goat anti-mouse antibody conjugated to Alexa-647 (1:2,000 dilution). Samples were analyzed using BD FACSVerse followed by FlowJo software (TreeStar Corp.).

## Nocodazole treatment

A549 cells were seeded and infected as described above. Three hours after initiation of infection, media was replaced with 30 µM Nocodazole (diluted in DMSO, #M1404, Sigma) or the

equivalent volume of DMSO in Opti-MEM. FlAsH staining was performed in 30 μM Nocodazole or equivalent volume of DMSO in Opti-MEM. FISH staining was performed as described above.

## Nuclear and cytoplasmic RNA extraction

A549 cells were seeded at a density of $8 \times 10^5$ cells per well in 6-well dishes and infected at MOI = 5 in triplicate with 3 wells for each replicate. At 7hpi, the RNA from cytoplasmic and nuclear fractions were isolated using a cytoplasmic & nuclear RNA extraction kit following the manufacturers recommendations (Norgen Biotek Corp., Canada). Enrichments for nuclear and cytoplasmic fractions were assessed by semiquantitative RT-PCR analysis for U6 and S14 RNA, respectively. RNA was reverse transcribed using SuperScript II reverse transcriptase (Invitrogen) using random hexamers (Invitrogen). PCR was performed using gene specific primers and EconoTaq PCR Master Mix (Lucigen) per the manufacturer's guidelines (18 cycles for S14 and 24 cycles for U6). PCR primers are as follows- S14: forward GGCAGACCG AGATGAATCCTC and reverse CAGGTCCAGGGGTCTTGGTCC; U6- foward GTGCTC GCTTCGGCAGCACATATAC and reverse AAAAATATGGAACGCTTCACGAATTTG. vRNA copy number estimation is described below.

## Estimation of non-infectious particles

PFU to HAU ratio: Virus particles in supernatants were measured by standard hemagglutination (HA) assay using serially diluted cell supernatants in 0.5% chicken red blood cells (RBC). HA units (HAU) were determined by the reciprocal of the highest dilution of virus supernatant positive for RBC agglutination. Infectious titers were determined by plaque assay in MDCK cells as described above. The PFU to HAU ratio was calculated for two independent experiments, each comprising three replicates.

Virion purification and vRNA copy number to PFU ratio: Rab11a KO and control A549 cells were seeded at a density of $1.2 \times 10^7$ cells in 15-cm dishes coated with Poly-L-Lysine a day prior. PR8 infection experiments were performed at an MOI of 0.01 as described above. At 72hpi, supernatants were collected, clarified of debris by low speed centrifugation at 3,000rpm for 10min and filtered using 0.45uM filtration units. Supernatants were layered onto a 25% sucrose solution (NTE buffer) and ultracentrifuged in a SW28 rotor at 25,000 rpm for 90min. Virion pellets were resuspended in NTE buffer at $1/100^{th}$ of initial volume of the supernatant. Viral RNA was isolated from resuspended virion pellets using the QIAaMP Viral RNA Mini Kit (Qiagen) and viral cDNA was generated using the universal F(A) influenza primer (GTTACGCGCCAGCAAAAGCAGG) and the Maxima Reverse Transcriptase (Thermo Scientific). RNA Copy Number was quantitatively determined by Droplet Digital qPCR (Bio-Rad) of viral cDNA to determine RNA Copy Number/ml for all 8 segments, using sequence specific primers (shown below). The RNA Copy Number/mL to PFU/mL or HAU ratio was calculated for three replicates.

## Western blot analysis

Cells were lysed in RIPA buffer (50mM Tris pH 7.4, 150mM NaCl, 0.1% SDS, 0.5% deoxy cholate, 1% Triton X-100) supplemented with protease inhibitors (Sigma), quantified, and protein samples were separated on a 10% SDS-PAGE gel. Western blot analysis was performed following the transfer of proteins onto a nitrocellulose membrane. Antibodies used were NP (#4862, BEI resources), Ku (#2882, Sigma), Rab11a (#715300, Invitrogen), and goat α-mouse or goat α-rabbit secondary antibodies conjugated to horseradish peroxidase (#NA934V and NA931V, Sigma).

| PR8 Primers | | | |
|---|---|---|---|
| PR8 PB2 309F | GGCTGTGACATGGTGGAATA | PR8 PB2 410R | CCATGCTTTAGCCTTTCGAC |
| PR8 PB1 393F | ACAAGGCCGACAGACCTATG | PR8 PB1 465R | TATTGTGTTGGCCAATGCTG |
| PR8 PA 363F | GGAGAATAGATTCATCGAAATTGG | PR8 PA 438R | AATTTTATTGGCCTTTTCCAGA |
| PR8 HA 381F | GAGCTGAGGGAGCAATTGAG | PR8 HA 474R | CTCCGTTTGTGTTGTGGTTG |
| PR8 NP 336F | CAGGAGAGTAAACGGAAAGTGG | PR8 NP 407R | CGCCAGATTCGCCTTATTT |
| PR8 NA 351F | CACTTGGAATGCAGGACCTT | PR8 NA 420R | CAGTCCCATTTGAATGCTTG |
| PR8 M 460F | CCTGGTATGTGCAACCTGTG | PR8 M 540R | AGTGGATTGGTGTTGTCACC |
| PR8 NS 480 F | GAAGAGGGAGCAATTGTTGG | PR8 NS 562 R | CCAACTGCATTTTTGACATCC |
| Pan 99 Primers | | | |
| P99 PB2 322F | TGGAATAGAAATGGACCTGTGA | P99 PB2 414R | GGTTCCATGTTTTAACCTTTCG |
| P99 PB1 508F | AGGCTAATAGATTTCCTCAAGGATG | P99 PB1 596R | ACTCTCCTTTTTCTTTGAAAGTGTG |
| P99 PA 307F | TGCAACACTACTGGAGCTGAG | P99 PA 398R | CTCCTTGTCACTCCAATTTCG |
| P99 HA 251F | CCTTGATGGAGAAAACTGCAC | P99 HA 313R | CAACAAAAAGGTCCCATTCC |
| P99 NP 482F | CAACATACCAGAGGACAAGAGC | P99 NP 571R | ACCTTCTAGGGAGGGTCGAG |
| P99 NA 386F | TCATGCGATCCTGACAAGTG | P99 NA 461R | TGTCATTTGAATGCCTGTTG |
| P99 M 563F | GTTTTGGCCAGCACTACAGC | P99 M 662R | CCATTTGCCTGGCCTGACTA |
| P99 NS 252F | ACCTGCTTCGCGATACATAAC | P99 NS 342R | AGGGGTCCTTCCACTTTTTG |

## Statistical analysis

Significance of data points was assessed using the unpaired Student's t-test. p-value < 0.05; [**] p-value < 0.01; [***] p-value < 0.001; N.S., non-significant. For the RNA Copy Number:PFU ratios, significance of data points was assessed using ordinary 2-way ANOVA.

## Supporting information

**S1 Fig. Comparison of vRNP dynamics between WT PR8 and NP-Tc virus.** (A-B) A549 cells were infected with WT PR8 or NP-Tc virus at MOI = 1 for the indicated time points and stained with FlAsH dye and an anti-RNP antibody. (C) Comparison of RNP puncta sizes between NP-Tc and WT PR8 viruses. (D) Comparison of number of RNP puncta in control A549 cells infected with NP-Tc and WT PR8 viruses. Scale bar = 10 μm. Data are representative of at least two independent experiments.
(TIF)

**S2 Fig. Large RNP puncta appear near Sec31a positive ER exit sites.** Control A549 cells were infected with NP-Tc virus and at 16hpi, cells were stained with FlAsH dye and a Sec31a antibody. RNP puncta with Sec31a in the periphery are indicated by arrows in the zoomed image.
(TIF)

**S3 Fig. Snap shots from live-cell imaging of NP-Tc virus infected cells.** Selected frames from S1 Movie are shown with time (min:sec). Arrows indicate different events: RNP accumulation (green), puncta splitting (yellow) or coalescing (blue).
(TIF)

**S4 Fig. Loss of Rab11a affects post replication steps of the IAV life-cycle.** (A-B) Rab11a KO cells show defects at a post replication step. (A) Control A549 and Rab11a KO cells were infected with H1N1-GFP (PR8; MOI = 3) and H5N1-GFP (VN04; MOI = 1) and GFP expression was assessed at 16hpi by flow cytometry. (B) Control A549 and Rab11a KO cells were infected with H1N1 (PB2-Gluc; PR8) at MOI = 2 or 10 and incubated in media without TPCK

trypsin to restrict infection to a single round. At 16hpi, luciferase activity in cell lysates was measured and is shown relative to mock lysates. Values are shown as relative light units (RLU). (C) Surface expression of viral proteins are not affected in Rab11a KO cells. Control A549 and Rab11a KO cells were infected with WT PR8 at MOI = 1 and surface expression of viral HA, NA and M2 were measured by flow cytometry. (D) Rab11a KO cells lack large FlAsH puncta. Control A549 and Rab11a KO cells were infected with NP-Tc virus at MOI = 1 for the indicated times and FlAsH staining as well as a-tubulin staining were performed. Scale bar = 10 μm. Data shown is a representative of two independent experiments performed with three biological replicates.
(TIF)

**S5 Fig. Semiquantitative RT-PCR analysis for the quality of nuclear and cytoplasmic fractions.** Control A549 and Rab11a KO cells were either mock (A) or WT PR8 infected (B), and at 7hpi, RNA from nuclear and cytoplasmic fractions were isolated. The quality of the fractions was assessed by semi-quantitative RT-PCR using S14 and U6 specific primers for 18 and 24 cycles, respectively. Data shown is a representative of two independent experiments performed with two biological replicates.
(TIF)

**S6 Fig. Absence of Rab11a increases the production of defective particles.** Rab11a KO cells show increased production of defective virions. (A-B) Control A549 and Rab11a KO cells were infected with H1N1(PR8) or H3N2 (Pan99) for 48 hours, and supernatants were tittered by plaque assay and subjected to viral RNA extraction. Viral RNAs were reverse transcribed and individual segment copy numbers were quantified by digital droplet PCR. The ratios of viral RNA copy number to PFU are shown for all eight segments of H1N1 and H3N2 viruses. (C) Viral titers at different stages of virion purification. Control A549 and Rab11a KO cells were infected with WT PR8 at MOI = 0.01 and at 72hpi, supernatants were collected and clarified of debris by low speed centrifugation. Subsequently, supernatants were filtered through a 0.45uM filter and pelleted on a 25% sucrose cushion. (D) Comparison of vRNA copy numbers in virions isolated from control A549 and Rab11a KO cells. $^*$ p-value $< 0.05$; $^{**}$ p-value $< 0.01$; $^{***}$ p-value $< 0.001$; N.S., non-significant. Data shown is a representative of two independent experiments performed with three biological replicates.
(TIF)

**S1 Movie. Live-cell imaging of NP-Tc infected cells.** A549 cells transduced with mCherry-Rab11a were infected with NP-Tc virus at MOI = 3. At 16hpi, cells were stained with FlAsH dye and live-imaging was performed for 30min with images captured at 20 sec intervals. FlAsH (green), mCherry-Rab11a (red) and nuclei (blue). RNP puncta splitting or coalescing are indicated by arrows at the indicated times.
(AVI)

## Acknowledgments

We would like to thank Dr. Adolfo Garcia-Sastre (Icahn School of Medicine) for sharing numerous reagents and Dr. Ron Fouchier (Erasmus Medical Center in Rotterdam) for providing the reverse genetics plasmids for H7N7 virus and Dr. Peter Palese for the PR8 PB2-Gluc plasmid. Also, we thank Shea Lowery for help with FISH staining. Imaging was performed at the Integrated Light Microscopy Core Facility at The University of Chicago, and we would like to thank Dr. Christine Labno for microscope training and her assistance in optimizing FISH imaging.

## Author Contributions

**Conceptualization:** Julianna Han, Jasmine T. Perez, Balaji Manicassamy.

**Data curation:** Julianna Han, Ketaki Ganti, Veeresh Kumar Sali, Carly Twigg, Yifeng Zhang.

**Formal analysis:** Julianna Han, Ketaki Ganti, Carly Twigg, Yifeng Zhang.

**Funding acquisition:** Lillianna Radoshevich, Anice C. Lowen, Balaji Manicassamy.

**Investigation:** Julianna Han, Ketaki Ganti, Veeresh Kumar Sali, Carly Twigg, Yifeng Zhang, Senthamizharasi Manivasagam, Chieh-Yu Liang, Olivia A. Vogel, Iris Huang, Shanan N. Emmanuel, Jesse Plung, Lillianna Radoshevich, Jasmine T. Perez, Anice C. Lowen, Balaji Manicassamy.

**Methodology:** Julianna Han, Ketaki Ganti, Jasmine T. Perez, Anice C. Lowen, Balaji Manicassamy.

**Project administration:** Balaji Manicassamy.

**Resources:** Lillianna Radoshevich, Anice C. Lowen, Balaji Manicassamy.

**Supervision:** Lillianna Radoshevich, Anice C. Lowen, Balaji Manicassamy.

**Validation:** Julianna Han, Ketaki Ganti, Carly Twigg, Senthamizharasi Manivasagam.

**Visualization:** Julianna Han, Jasmine T. Perez, Balaji Manicassamy.

**Writing – original draft:** Julianna Han, Jasmine T. Perez, Balaji Manicassamy.

**Writing – review & editing:** Julianna Han, Ketaki Ganti, Jasmine T. Perez, Anice C. Lowen, Balaji Manicassamy.

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
