## [Decision Letter · Decision Letter 0]

19 Apr 2021

Dear Dr. Manicassamy,

We are pleased to inform you that your manuscript 'Host Factor Rab11a is Critical for Efficient Assembly of Influenza A Virus Genomic Segments' has been provisionally accepted for publication in PLOS Pathogens.

Best regards,

Andrew Pekosz, Ph.D.

Section Editor

PLOS Pathogens

Andrew Pekosz

Section Editor

PLOS Pathogens

Kasturi Haldar

Editor-in-Chief

PLOS Pathogens

orcid.org/0000-0001-5065-158X

Michael Malim

Editor-in-Chief

PLOS Pathogens

orcid.org/0000-0002-7699-2064

Reviewer Comments (if any, and for reference):

Reviewer's Responses to Questions

**Part I - Summary**

Reviewer #1: The paper by Han et al, entitled “ Host Factor Rab11 is Critical for Efficient Assembly of Influenza A Virus Genomic Segments” builds on previously published data, from different groups, strongly suggesting the involvement of the cellular GTPase Rab11 in trafficking of the influenza vRNPs. From the nucleus to the budding site at the cellular membrane.

The manuscript describes two new tools to investigate the role of Rab11, a new virus where NP is tagged at the C-terminus with a small TC-tag and Rab11 KO cells. The conclusions reached by the authors are in line with previous data and are, as such, not new. The manuscript however enlarges these conclusions to a broad range of influenza viruses and reveals the interesting fact that viruses produced from Rab11 KO cells have a higher rate of non-infectious particles, as measured by the ratio of RNA copy number to PFU.

I have been invited to review this paper after an initial round of substantial revision and could therefore see the comments of the initial reviewers and the responses of the authors to these comments.

I consider that the initial reviews were extremely thorough and that the authors have been extremely careful at responding very convincingly to almost all the criticisms and suggestions of the reviewers. The authors have carried out many new experiments, in a very appropriate manner.

Hence, even though I agree with the reviewers on the fact that the data mainly corroborate exiting data, there are interesting additions to them that, together with the new tools described in the paper, make the manuscript suitable for publication.

Some more questions/suggestions to the authors:

Their NP-Tc virus carries two packaging sequences. It would probably be more suitable to degenerate the original packaging sequence that precedes the Tc-tag, while preserving the coding sequence, in order to avoid strict duplication of the packaging signals that could interfere with the packaging process. May be such a virus would become more similar to the WT in terms of fitness?

I am also wondering why the authors chose a chicken PolI promoter for WSN minireplicon assay in 293T cells?

And finally, looking at viral particles budding from WT or Rab-11 KO cells by electron microscopy could be an elegant addition to the RT-qPCR quantification of vRNAs from sucrose purified viral particles. Indeed, it would enable to see whether Rab11 KO cells produce more empty particles, or particles with less than 8 vRNPs (although one vRNP could be replaced by rRNA), compared to WT cells.

Reviewer #2: The revised version has addressed the concerns of the reviewers and the manuscript is, in my opinion, relevant to the scientific community as it increases our understanding of how influenza A virus epidemic and pandemic genomes assemble. To my knowledge, Rab11a, emerges as the first host factor that negatively impacts influenza A virus selective process of genome assembly into a 8-partite complex. The paper includes two important novelties: Rab11a reduces influenza A virus production because it reduces the efficiency in genome assembly and this is evaluated in Rab11a knocked-out cells. The authors have satisfied the doubts of this reviewer. Two note, it would be interesting to see if in Rab11a knock-out cells vRNPs associate with the ER and the discussion is very much a summary of the entire paper and not exactly a discussion of the implications of the findings for the global understanding of the viral lifecycle.

**Part II – Major Issues: Key Experiments Required for Acceptance**

Reviewer #1: (No Response)

Reviewer #2: No major issues found

**Part III – Minor Issues: Editorial and Data Presentation Modifications**

Reviewer #1: (No Response)

Reviewer #2: No minor issues found

PLOS authors have the option to publish the peer review history of their article (what does this mean?). If published, this will include your full peer review and any attached files.

Reviewer #1: No

Reviewer #2: No

---

## [Editor Report · Acceptance letter]

6 May 2021

Dear Dr. Manicassamy,

We are delighted to inform you that your manuscript, "Host Factor Rab11a is Critical for Efficient Assembly of Influenza A Virus Genomic Segments," has been formally accepted for publication in PLOS Pathogens.

Best regards,

Kasturi Haldar

Editor-in-Chief

PLOS Pathogens

orcid.org/0000-0001-5065-158X

Michael Malim

Editor-in-Chief

PLOS Pathogens

orcid.org/0000-0002-7699-2064